# Restrained Eating Is Associated with Lower Cortical Thickness in the Inferior Frontal Gyrus in Adolescents

**DOI:** 10.3390/brainsci11080978

**Published:** 2021-07-23

**Authors:** Isabel García-García, Maite Garolera, Jonatan Ottino-González, Xavier Prats-Soteras, Anna Prunell-Castañé, María Ángeles Jurado

**Affiliations:** 1Department of Clinical Psychology and Psychobiology, University of Barcelona, 08035 Barcelona, Spain; isabel.garciagarcia@ub.edu (I.G.-G.); jottinog@uvm.edu (J.O.-G.); x.prats.soteras@gmail.com (X.P.-S.); aprunell@ub.edu (A.P.-C.); 2Neuropsychology Unit, Hospital of Terrassa, Consorci Sanitari de Terrassa, 08227 Terrassa, Spain; mgarolera@cst.cat; 3Department of Psychiatry, University of Vermont College of Medicine, Burlington, VT 05405, USA; 4Institut de Neurociències, University of Barcelona, 08035 Barcelona, Spain; 5Institut de Recerca Sant Joan de Déu, Hospital Sant Joan de Déu, 08950 Esplugues de Llobregat, Spain

**Keywords:** restrictive eating, dieting, uncontrolled eating, binge eating, executive functions, personality, impulsivity, eating disorders, cortical thickness, subcortical volume

## Abstract

Some eating patterns, such as restrained eating and uncontrolled eating, are risk factors for eating disorders. However, it is not yet clear whether they are associated with neurocognitive differences. In the current study, we analyzed whether eating patterns can be used to classify participants into meaningful clusters, and we examined whether there are neurocognitive differences between the clusters. Adolescents (*n* = 108; 12 to 17 years old) and adults (*n* = 175, 18 to 40 years old) completed the Three Factor Eating Questionnaire, which was used to classify participants according to their eating profile using k means clustering. Participants also completed personality questionnaires and a neuropsychological examination. A subsample of participants underwent a brain MRI acquisition. In both samples, we obtained a cluster characterized by high uncontrolled eating patterns, a cluster with high scores in restrictive eating, and a cluster with low scores in problematic eating behaviors. The clusters were equivalent with regards to personality and performance in executive functions. In adolescents, the cluster with high restrictive eating showed lower cortical thickness in the inferior frontal gyrus compared to the other two clusters. We hypothesize that this difference in cortical thickness represents an adaptive neural mechanism that facilitates inhibition processes.

## 1. Introduction

Eating disorders, such as anorexia nervosa, bulimia nervosa, and binge eating disorder, are common mental health problems. For instance, in the United States, their life-time prevalence is estimated to be around 0.80%, 0.28%, and 0.85%, respectively [1]. Independent studies have identified that some eating patterns constitute risk factors for the development of eating disorders [2]. For example, restrictive eating patterns, also referred to as dieting, seem to precede the onset of both anorexia nervosa and bulimia nervosa in some individuals [2,3,4]. Along similar lines, uncontrolled eating, which might be referred to as overeating, seems to predict the onset of bulimia nervosa and binge eating disorder [3,4]. A focus on non-clinical eating behaviors might thus represent a valuable opportunity for nutrition studies since they can offer new insights about the subclinical stages of eating disorders.

Cluster analysis facilitates a way to characterize eating behaviors by classifying participants into subgroups according to their eating profiles. Using self-reported data on eating tendencies, previous studies have found at least two clusters that signal potentially problematic eating patterns: a restrictive eating cluster (in one publication this group was referred to as “food avoidance”) and a cluster with high scores in uncontrolled eating [5,6,7]. It is currently not known, however, if these clusters are kept stable across different age groups. This might be particularly relevant, since it is known that the majority of patients with anorexia and bulimia nervosa show an early onset of the disease, i.e., before the patients reach the age of 22 years [8].

Previous work in cognitive neuroscience has examined the relationship between eating behaviors, brain differences in structure and function, and cognitive function. Eating patterns that are potentially problematic, such as restrictive eating or uncontrolled eating, seem to be modulated by at least three brain circuits: the mesolimbic circuit, the cognitive control network and the stress system [9].

The mesolimbic circuit processes the motivational and rewarding aspects of food and is key in subjective valuation [10]. It arises from midbrain dopamine signals projected to the striatum, and it includes other limbic areas connected to the midbrain or to the striatum, such as the orbitofrontal cortex, thalamus, hippocampus, and amygdala [10,11]. Alterations in the mesolimbic network could potentially trigger impulsive eating by increasing the incentive salience of food [12]. The cognitive control network might be especially relevant in sustaining restrictive eating patterns. Functional MRI studies have shown that, when participants exert dietary self-control, a set of brain regions that conform the cognitive control network become activated. Some of these regions are the anterior insula, inferior frontal gyrus/ventrolateral prefrontal cortex, dorsolateral prefrontal cortex, and temporal–parietal junction [13]. Finally, stress is known to induce marked changes in eating behaviors. Upon the detection of an acute stressor, the hypothalamus releases the corticotropin-releasing hormone, provoking the activation of the hypothalamus–pituitary–adrenal (HPA) axis. Corticotropin-releasing hormone, moreover, is known to act as an appetite suppressor [14]. However, chronic stress is known to stimulate appetite [14] and, for some people, it can lead to unhealthy weight gain [15]. In addition to the HPA axis, two medial temporal regions are also important for stress responses: the amygdala and the hippocampus. The amygdala has a crucial role in salience detection [16] and is robustly engaged during the visualization of negatively-valenced stimuli [17]. Its projections via the sensory cortex might enable the organism to increase attentional processes towards stressful stimuli [18]. Potentially, these three circuits could be involved in different manifestations of a given eating disorder (for example, in the case of patients with the binge purge subtype of anorexia nervosa, the three circuits could hypothetically play a role in different symptoms).

Neuroimaging studies have examined the associations between eating patterns and neuroanatomical differences. Initial evidence, however, is scarce and heterogeneous. High scores in restrictive eating seem to be associated with lower gray matter volume in the putamen nucleus, along with higher gray matter volume in the dorsolateral prefrontal cortex [19]. Another study found that restrictive eating was correlated with lower gray matter volume in the precuneus [20]. On the other hand, higher scores in uncontrolled eating have been associated with increases in gray matter volume in the nucleus accumbens [21]. In contrast, other studies have found uncontrolled eating to be related with decreases in gray matter volume in the dorsolateral prefrontal cortex [19] and orbitofrontal cortex [22].

It is possible that some of these neuroanatomical differences associated with eating patterns, in turn, relate to differences in cognitive function. In that regard, some studies have examined the association between eating patterns and executive functions. In two studies, restrictive eating was not related to executive function performance [23,24]. However, another study reported that, in participants with lower scores in executive functions, restrained eating was correlated with higher sensitivity to punishment and higher reward responsivity [25]. With regards to uncontrolled eating patterns, we found null associations between this eating trait and performance in executive functions across 10 independent studies [26]. One possible explanation for this is that eating traits might be associated with specific subdomains of executive function performance (such as working memory, cognitive flexibility, or planning), rather than with general executive function deficits. An extensive evaluation of different subdomains of executive functions in relation with non-clinical eating traits is, however, missing.

The current study has two main goals. First, we seek to explore whether we can classify participants into subtypes based on their eating patterns. Specifically, we will examine participants classified in two age groups: adolescence and young adulthood. Second, we aim to define the neurocognitive profile of each eating behavior cluster. To this aim, we will test for differences in regions of the brain belonging to the mesolimbic, cognitive control or stress circuits of the brain, as well as for differences in executive functions and personality across clusters.

## 2. Materials and Methods

### 2.1. Participants

Participants were recruited as part of two broader projects aimed at identifying the neuro-behavioral correlates of overweight and obesity in adolescence [27] and adulthood [28,29]. Participants gave their informed consent following the Helsinki′s declaration, and the Committee of Bioethics of the University of Barcelona approved the projects. The age of the participants was between 12 and 40 years old, and they were split into two independent samples: (i) adolescents (*n* = 108; age range from 12 to 17 years old), and (ii) adults (*n* = 175; aged 18 to 40 years old). This classification was done by considering that adulthood starts at the same time as the most common voting age. Exclusion criteria were clinical history of psychiatric illness (including addictive and/or eating disorders), and clinical history of developmental, neurological, or systemic diseases (such as hyper- or hypothyroidism, diabetes, or cardiovascular diseases). Being claustrophobic and being a bearer of non-removable metallic objects (such as orthodontics) were additional exclusion criteria for the MRI.

### 2.2. Eating Behavior Scales

Eating patterns were examined using the Three Factor Eating Questionnaire (TFEQ) [30] and the Bulimia Inventory Test of Edinburgh (BITE) [31].

The TFEQ was administered in its 18 items revised form, which provides three measures of human eating behavior: cognitive restraint, or the tendency to suppress food intake, uncontrolled eating, or the tendency to feel losses of control over food intake, and emotional eating, or the tendency to eat in response to negative emotional states. The scores of the TFEQ were used in the cluster analysis to define different subgroups of participants.

The BITE test provides a symptom score reflecting binge eating and bulimic tendencies. Since none of the participants presented purging symptoms, high scores in this test should be interpreted as high binge eating tendencies (similar to the uncontrolled eating scale from the TFEQ). Scores were used to test for the validity of the cluster classification.

### 2.3. Clinical Variables

We measured weight, height, and waist circumference in order to calculate body mass index (BMI) and waist-to-height ratio. The waist-to-height ratio was used as an indicator of central adiposity, since it is considered a strong predictor of type 2 diabetes and cardiovascular disorders, in adults [32] as well as in pediatric populations [33]. BMI was additionally used to categorize participants in obese, overweight, and lean groups, following the standard categorization of the World Health Organization. Moreover, in the adolescent sample, age and sex were taken into account in the definition of obese, overweigh, and lean groups, as proposed by Cole et al. [34]. We tested whether results changed if groups reflecting body-weight status were entered in the analyses instead of waist-to-height ratio, but the significant results obtained remained the same.

Participants completed the Hospital Anxiety and Depression Scale (HADS) [35], and the total sum of items was computed.

### 2.4. Executive Functions

We used the following neuropsychological tests to examine executive function performance. We administered a computerized version of the Wisconsin card sorting test (WCST) [36] and recorded the total number of errors, and the Trail Making Test part B minus part A [37]. These tests are generally considered to measure cognitive flexibility [38]. To assess working memory [38], we administered letter-number sequencing from the Wechsler Intelligence Scales (WISC or WAIS). Finally, to evaluate inhibitory control [38], we recorded the interference score from the Stroop test [39], and number of commission errors from the Continuous Performance Test, 2nd Edition (CPT-II) [40].

### 2.5. Personality

We administered the Temperament and Character Inventory Revised (TCI-R). The TCI-R evaluates personality based on Cloninger’s psychobiological model of personality. It provides measures for four temperament dimensions: novelty seeking, harm avoidance, reward dependence and persistence; along with three character dimensions: self-directedness, cooperativeness and self-transcendence [41,42].

As a measure of impulsivity, we used a self-report delay discounting questionnaire [43], which evaluates the tendency to choose immediate small rewards over large delayed ones.

We additionally administered the Barratt Impulsivity Scale 11 (BIS-11) to the adult sample, which provides three subscales of impulsivity: cognitive, motor, and non-planned.

Finally, we also evaluated self-esteem in adolescents with the Rosenberg’s self-esteem scale.

### 2.6. MRI Acquisition

Participants were asked to participate in a brain MRI acquisition, performed on a separate day. A number of participants did not perform this session, which reduced the adolescent MRI subsample to *n* = 60 and the adult MRI subsample to *n* = 106. The main reasons why participants dropped out of the MRI sub-study were schedule incompatibilities and meeting exclusion criteria for the MRI (see Section 2.1.).

We acquired a high resolution T1-weighted 3D using an MPRAGE echo sequence with the following parameters: repetition time 2300 ms., echo time 2.98 ms., inversion time 900 ms. We acquired 2401 mm contiguous slices using a 256 × 256 matrix with an in-plane resolution of 1 × 1 mm^2^.

### 2.7. MRI Processing: Cortical and Subcortical Brain Regions

We performed the preprocessing and analysis of cortical thickness and subcortical volumes using FreeSurfer software (Version 6.0; https://surfer.nmr.mgh.harvard.edu (accessed on 1 May 2016)). The process included volume correction and average of T1, removal of non-brain tissue, intensity normalization, and tessellation of gray/white matter tissue. After the processing, we performed visual inspection to ensure accuracy of registration, skull stripping, segmentation, and cortical surface reconstruction. Data resulting from this processing has been published in two studies examining the association between obesity and brain morphometry in adolescence [27] and adulthood [29].

We extracted mean subcortical volume of the following regions of interest, by using the Desikan atlas: caudate, putamen, accumbens, pallidum, amygdala and hippocampus. Data from the left and right side of the brain were added, and the value obtained was divided by total subcortical gray matter volume.

Additionally, we extracted mean thickness values from the following cortical structures: orbitofrontal cortex (medial and lateral), inferior frontal gyrus (opercularis, orbitalis, and triangularis), insula, and anterior cingulate cortex (caudal and rostral). We divided the values obtained by global cortical thickness.

These regions were chosen because of their involvement in the mesolimbic circuitry (e.g., caudate, putamen, accumbens, pallidum, amygdala, insula, and orbitofrontal cortex), in the cognitive control circuit (e.g., inferior frontal gyrus, insula, and anterior cingulate cortex), as well as in the stress system (e.g., hippocampus and amygdala).

### 2.8. Cluster Analysis

The three scales of the TFEQ were centered and entered into a k-means cluster analysis using the “cluster” package (Version 2.1.0.) available in R (Version 4.0.0.) [44]. In both samples, we chose k = 3 since this number allowed us to keep a relatively large sample in each cluster. This decision was in congruence with the elbow method. More specifically, for a range of different values of k (2 to 15) we calculated the total within-cluster sum of square (WSS) and plotted it (Appendix A). For both samples, k values superior to 3 did not substantially improve the WSS value.

To examine the meaningfulness of the cluster classification, we tested for cluster differences in the BITE test (extra-cluster validation).

### 2.9. Minimally- and Fully-Adjusted ANCOVA Models

We analyzed the effect of cluster classification on executive functions, personality variables, and brain anatomy. To do so, we built minimally adjusted ANCOVA models that included cluster classification as main covariate of interest, along with those demographic and clinical variables that showed statistical differences between clusters.

In adolescents, clusters showed differences in waist-to-height ratio (Table 1), so we included this variable in the minimally adjusted models. In adults, clusters differed in terms of waist-to-height ratio, sex distribution, and anxiety/depression symptoms (see Table 2). These variables were then included in the minimally adjusted models.

When a significant effect of cluster classification was found, we additionally built a fully adjusted ANCOVA model that accounted for the possible effects of demographic and clinical variables. In both samples, the fully adjusted ANCOVA models included cluster classification along with age, sex distribution, waist-to-height ratio, and anxiety/depression symptoms.

We adjusted our significant level threshold to account for multiple testing using Bonferroni correction. In adolescents, we calculated 23 models (i.e., five models examining executive functions, nine models testing personality, and nine models examining neuroanatomical regions of interest). The significance level in this sample was set to *p* < 0.002 (0.05/23). In adults, we performed 25 models (i.e., five models examining executive functions, 11 models testing personality, and nine models examining brain regions of interest), so the significant *p* value was set to *p* < 0.002 (0.05/25).

## 3. Results

### 3.1. Adolescent Sample (n = 108)

#### 3.1.1. Cluster Analysis

The three cluster solution provided a cluster characterized by high scores in uncontrolled eating and emotional eating (“uncontrolled eating”), a second group with high scores in cognitive restraint (“restrained eating”), and a third cluster with low scores in the three eating scales (“low problematic eating”) (Figure 1).

The three clusters differed from each other with regards to the scales of the TFEQ (intra-cluster variables), and with regards to binge _eating_ symptoms (extra-cluster variable). This suggests that the segregation of participants into three groups was meaningful. The three clusters differed with regards to waist-to-height ratio, with no differences in the other demographic and clinical variables examined (i.e., age, sex distribution, anxiety and depression symptoms) (see Table 1). The minimally-adjusted ANCOVA models were then corrected for waist-to-height ratio, while age, sex, and anxiety/depression symptoms were included in the fully-adjusted ANCOVA models.

#### 3.1.2. Executive Functions and Personality

There were no significant differences between the three groups in executive functions, nor there was a cluster effect in any of the personality measurements evaluated (Appendix A).

#### 3.1.3. Neuroanatomical Results (Subsample: *n* = 60)

In the neuroimaging subsample, the proportion of participants was distributed as follows: uncontrolled eating *n* = 19, restrained eating *n* = 15, low problematic eating *n* = 26. Clinical characteristics of the neuroimaging subsample are shown in the Appendix A (Appendix A). We found a cluster effect in cortical thickness of the inferior frontal gyrus. This effect was found both in the minimally-adjusted model (F = 8.855; Bonferroni-corrected *p* = 0.011), which controlled for waist-to-height ratio, as well as in the fully adjusted model (F = 8.574; Bonferroni-corrected *p* = 0.014), which controlled for waist-to-height ratio, age, sex distribution, and anxiety/depression symptoms. Compared to the low problematic eating cluster, participants in the restrained eating cluster showed lower cortical thickness in the inferior frontal gyrus (Bonferroni-corrected *p* = 0.013) (Figure 2).

### 3.2. Adult Sample (n = 175)

#### 3.2.1. Cluster Analysis

The three cluster solution provided a cluster with highs scores in uncontrolled eating and emotional eating (“uncontrolled eating”), a group characterized by high scores in cognitive restraint (“restrained eating”), and a group of participants with low scores in the three eating scales (“low problematic eating”) (Figure 3).

This cluster solution provided a meaningful separation of participants, since the three groups differed from each other with regards to the TFEQ scales (intra-cluster scales), and also with regards to binge eating symptoms from the BITE test (extra-cluster variable). The three clusters differed with regards to the three obesity-related variables (waist-to-height ratio, BMI, and obesity status), anxiety and depression symptoms and sex distribution. These variables were then selected as covariates in the minimally adjusted ANCOVA models (Table 1).

#### 3.2.2. Executive Functions and Personality

There were no significant differences between the three groups in executive functions (Appendix A), nor there was a cluster effect on any of the personality measurements evaluated (TCI-R and BIS-11).

#### 3.2.3. Neuroanatomical Results (Subsample *n* = 106)

In the neuroimaging subsample, the proportion of participants was distributed as follows: uncontrolled eating *n* = 32, restrained eating *n* = 42, low problematic eating *n* = 32. The Appendix A (Appendix A) displays clinical characteristics of this neuroimaging subsample. We did not find group differences in any of the subcortical and cortical regions examined.

## 4. Discussion

In this study, we examined whether eating patterns can be used to group participants into meaningful clusters. To do so, we analyzed two independent samples divided by age: participants in adolescence (aged 12 to 17 years) and adult participants (aged 18 to 40 years old). In both samples, we obtained three subtypes of participants: a group characterized by uncontrolled eating behavior, a cluster characterized by restrained eating, and a cluster with low scores in problematic eating. Next, we sought to define the neurobehavioral profile associated with each of these clusters. Clusters were similar with regards to cognitive function and personality profile across the two samples. In adolescents, there was a group effect of cortical thickness in the inferior frontal gyrus, in which participants in the restrained eating group showed lower thickness than both the uncontrolled eating and the low problematic eating groups. This difference was not observed in adults.

Eating patterns such as uncontrolled eating and restrictive eating constitute well-known risk factors for the development of eating disorders [2,3,4]. In the present study, we found that across two samples of non-clinical participants we obtained a cluster of participants showing high scores in uncontrolled eating patterns, and a cluster of participants showing high scores in restrained eating. The result suggests that the classification is relatively robust, and seems to show a certain stability from adolescence to middle adulthood.

The three clusters did not differ in terms of executive functions and personality. In a previous meta-analysis we showed that non-clinical uncontrolled eating patterns seem unrelated to performance in executive functions [26]. This stands in clear contrast with findings from clinical eating disorders, where different studies have shown that these patients perform worse than healthy controls in executive functions [26,45,46,47] (but see [48] for negative findings). The magnitude of these effects ranges from small, in the case of binge eating disorder [26], to medium size, in the case of anorexia nervosa and bulimia nervosa [45]. Together with our current finding, these results support the idea that non-clinical variations in eating behaviors are not accompanied by any conclusive differences in cognitive processing. Cognitive problems, however, might be observed in clinical populations (i.e., in patients with eating disorders), or in cases where the severity of the problematic eating patterns significantly interferes with daily activities.

In adults, there was a small tendency for participants with restrained eating behavior to show more symptoms of anxiety and depression. This effect, however, was not detected in the adolescent sample. This small difference might suggest that, taken to the extreme, restrained eating (or “dieting”) can potentially lead to anorexia nervosa [3], a psychiatric diagnosis with a frequent association with emotional symptoms [49].

With regards to neuroanatomical differences, in the adult sample there was no significant difference across clusters in the regions of interest examined. In the adolescent sample, however, participants classified into the group of restrictive eating showed lower cortical thickness in the inferior frontal gyrus. This effect was found while controlling for potential confounding factors, such as age, sex, waist-to-height ratio, and anxiety/depression symptoms. Brain cortical development during adolescence is characterized by an accelerated thinning with increasing age, which seems to be especially prominent in frontal lobe regions [50]. Hypothetically speaking, restrictive eating behaviors could interact with the neural development in the inferior frontal gyrus, a region associated with general inhibitory processes [51] and, more specifically, with dietary self-control [13]. This is purely speculative, since we would need longitudinal designs to trace this interaction. Moreover, we strongly argue against considering this possible variation of normal brain development as pathological. This is particularly so since the three groups showed similar scores in cognitive and personality measures. Alternatively, we suggest that it might reflect an adaptive neural mechanism that facilitates self-control processes in these participants.

In the following, we outline several limitations of the study and we suggest future research directions. First, the neuropsychological tests administered here were standard cognitive tests that have been widely used to evaluate executive functions [38]. However, these tests might not necessarily detect differences in executive functions that are specifically linked with food stimuli. For instance, participants with uncontrolled eating might show lower inhibitory control in response to food items, but that this might not necessarily translate to their performance in the classic Stroop or CPT-II tests. For this reason, the inclusion of food-related measures of executive functions would have been desirable and is a future line of research. Second, sex differences might have an influence on the neurobehavioral correlates of eating patterns. Pathological eating patterns seem to affect females and males differently. For instance, the prevalence of anorexia nervosa and bulimia nervosa is much higher in females than in males (e.g., [1]). While the current sample sizes were not robust enough to test for sex effects, future studies could examine if sex has an influence on the cognitive and personality profile of participants showing high scores in restrictive or uncontrolled eating. Finally, it would be interesting to investigate how changes in affective symptoms (such as anxiety and depression) might appear before or after certain eating behaviors, and to evaluate the neurocognitive correlates of affective symptomatology in clinical and subclinical eating disorders.

## 5. Conclusions

In this study we provide a comprehensive examination of the neurobehavioral correlates of non-clinical eating patterns in two samples: an adolescent sample (12–17 years old) and an adult sample (18 to 40 years old). To this aim, we classified participants into groups by using cluster analysis and obtained three clusters: a group characterized by high uncontrolled eating patterns, a group with high restrictive eating behavior, and a group with low problematic eating behavior. The three groups were equivalent in executive function performance and personality differences. In the adolescent sample, we found that participants with high restrictive eating behavior showed lower cortical thickness in the inferior frontal gyrus. There were no neuroanatomical differences in the adult sample. We suggest that the differences observed might reflect a greater engagement of self-control mechanisms in these participants.

## Figures and Tables

**Figure 1 brainsci-11-00978-f001:**
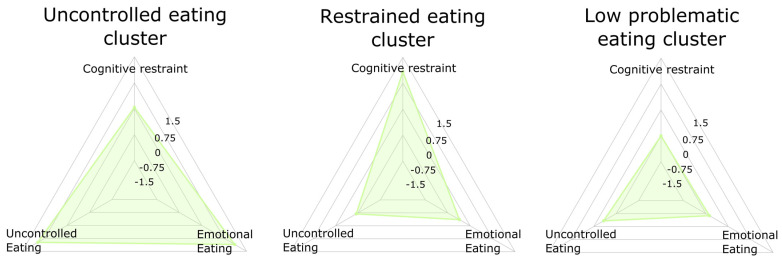
Radar charts depicting the three clusters obtained in the adolescent sample.

**Figure 2 brainsci-11-00978-f002:**
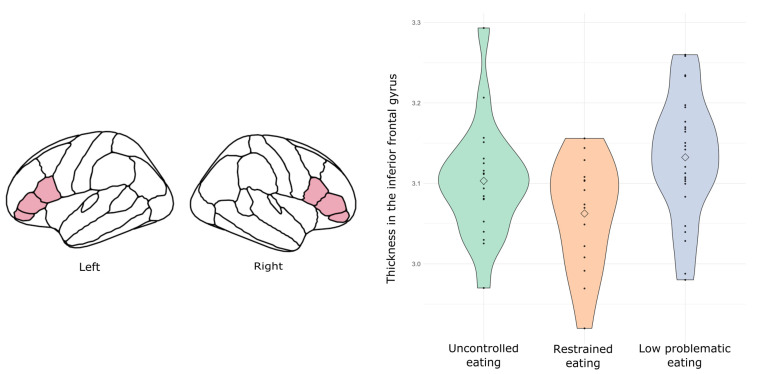
On the left, location of the inferior frontal gyrus, calculated as the sum of the three subregions provided by the Desikan atlas: pars opercularis, pars orbitalis, and pars triangularis (all depicted in pink). On the right, the violin plots show cortical thickness in the inferior frontal gyrus, obtained by each cluster.

**Figure 3 brainsci-11-00978-f003:**
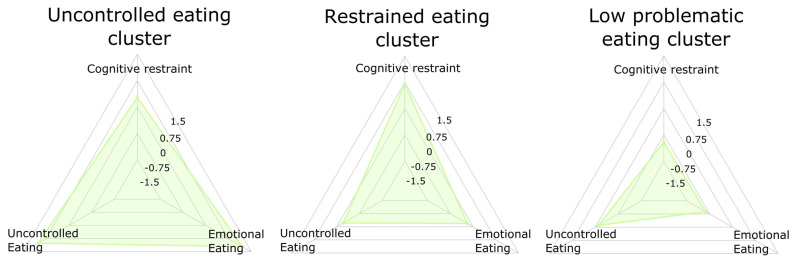
Radar charts representing the three clusters obtained in the adult sample.

**Table 1 brainsci-11-00978-t001:** Description of eating patterns, demographical variables, and clinical characteristics of the three clusters in the adolescent sample.

Domains	Variables	Uncontrolled Eating (*n* = 35)	Restrained Eating (*n* = 29)	Low Problematic Eating (*n* = 44)	F	*p*
Intra-cluster variables (TFEQ; centered)	Cognitive restraint	0.04 (0.87)	1.08 (0.57)	−0.75 (0.56)	63.54	<0.001
	Disinhibited eating	0.98 (0.77)	−0.69 (0.68)	−0.33 (0.70)	51.15	<0.001
	Emotional eating	1.10 (0.84)	−0.36 (0.64)	−0.63 (0.43)	76.71	<0.001
Extra-cluster validation (BITE)	BITE symptoms	6.65 (4.26)	2.71 (2.19)	2.66 (2.07)	19.93	<0.001
Demographic and clinical variables	Age	14.63 (1.70)	14.27 (1.50)	14.28 (1.60)	0.55	0.556
	Sex	19 females (54%) 16 males (46%)	14 females (48%) 15 males (52%)	21 females (48%) 23 males (52%)	X^2^(2) = 0.3825	0.826
	Waist (cm)/Height (cm)	0.53 (0.08)	0.55 (0.09)	0.48 (0.08)	12.17	<0.001
	BMI	28.11 (5.91)	28.82 (5.47)	23.5 (5.59)	9.60	<0.001
	Body weight status	10 lean (29%) 5 overweight (14%) 20 obese (57%)	5 lean (17%) 4 overweight (14%) 20 obese (69%)	26 lean (59%) 7 overweight (16%) 11 obese (25%)	X^2^(2) = 17.57	0.002
	Anxiety and depression (HADS)	8.18 (4.23)	7.34 (4.24)	6.31 (4.35)	1.81	0.170

**Table 2 brainsci-11-00978-t002:** Description of eating patterns, demographical variables and clinical characteristics of the three clusters in the adult sample.

Domains	Variables	Uncontrolled Eating (*n* = 54)	Restrained Eating (*n* = 61)	Low Problematic Eating (*n* = 60)	F	*p*
Intra-cluster variables (TFEQ; centered)	Cognitive restraint	0.27 (0.88)	0.70 (0.68)	−0.96 (0.53)	89.18	<0.001
	Disinhibited eating	1.01 (0.88)	−0.22 (0.47)	−0.10 (0.75)	85.22	<0.001
	Emotional eating	1.20 (0.62)	−0.19 (0.55)	−0.88 (0.39)	227.4	<0.001
Extra-cluster validation (BITE)	BITE symptoms	10.45 (5.80)	4.77 (3.29)	2.49 (2.61)	55.45	<0.001
Demographic and clinical variables	Age	31.15 (7.60)	30.69 (9.61)	30.67 (18.14)	0.06	0.945
	Sex	37 females (68.5%) 17 males (31.5%)	39 females (64%) 22 males (36%)	28 females (47%) 32 males (53%)	X^2^ = 6.42	0.040
	Waist (cm)/Height (cm)	0.60 (0.11)	0.56 (0.10)	0.51 (0.10)	12.17	<0.001
	BMI	31.60 (7.68)	29.80 (7.15)	25.81 (7.74)	8.97	<0.001
	Body weight status	12 lean (22%) 9 overweight (17%) 33 obese (61%)	16 lean (26%) 24 overweight (39%) 21 obese (34%)	41 lean (68%) 5 overweight (23%) 14 obese (8%)	X^2^ = 44.68	<0.001
	Anxiety and depression (HADS)	7.54 (4.79)	5.39 (3.55)	5.80 (3.78)	7.19	0.001

## Data Availability

To protect participants’ sensitive information, data is not publicly available, but it will be shared upon reasonable request.

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
