# Peer review of "Restrained Eating Is Associated with Lower Cortical Thickness in the Inferior Frontal Gyrus in Adolescents"

_brainsci, 2021, doi:10.3390/brainsci11080978_

Round 1

Reviewer 1 Report

The manuscript entitles “Restrained eating is associated with lower cortical thickness in the inferior frontal gyrus in adolescents” aimed to understand eating associated alteration in the cortical area.

In this study, the number of the sample is appreciated. More than 100 adolescents and adult subjects used. The variations between the groups show significant. The proper methods were performed to acquire the data and show high significant values between the groups. The female and male comparison is appreciated. The results and discussion are well written.

Author Response

We thank the reviewer for the positive comments. With regards to possible sex differences in the results, and also in response to another comment from Reviewer #2, we have added the following text in the Limitations/Future directions paragraph of the Discussion section:

TEXT ADDED:

Sex differences might have an influence on the neurobehavioral correlates of eating patterns. In fact, pathological eating patterns seem to affect females and males differently. For instance, the prevalence of anorexia nervosa and bulimia nervosa is much higher in females than in males [1]. While the current sample sizes were not robust enough to test for sex effects, future studies could examine if sex has an influence on the cognitive and personality profile of participants showing high scores in restrictive or uncontrolled eating.

Reviewer 2 Report

In this paper, authors explored the link between eating patterns and cortical thinness in both, adolescent and adults. Using a robust and common clustering algorithm, authors defined 3 food behavioral pattern and compared MRI data and neurocognitive outcomes. They found that, only in adolescent, highly restrictive eating behavior was associated with lower cortical thickness in the inf frontal gyrus. No difference was found regarding neurocognitive outcomes and personality.Results are of great interest, the methodology is accurate and the article is well written. Moreover, the authors avoid shortcuts and excessive inferences, especially regarding the links between food patterns and brain anatomy. 

The article meets the standards required for publication, without necessarily needing major revision. However, I have a few comments and queries:
-    Due to the use of data from studies related to obesity, some confusion may be present. If possible, authors may consider to better describe BMI, BMI ranges, percentage for each BMI categories?
-    Maybe sex and gender are under-explored, however, dietary patterns and cognitive functions are likely to be influenced by these variables. Perhaps for this study or a future one, gender-based or sex-based analysis could be considered?

Other comment : 
Line 31 :  maybe specify that the prevalence here is for US sample. This can differ in other studies.
Line 53-73 : the overall paragraph is well written and of interest. However, maybe authors could add that the three circuits could be altered in the different food patterns (maybe at different levels) (e.g., anorexia nervosa with food addiction) 
Line 113-116:  is there any exclusion criteria related to MRI technic? 
Line 135 : Even if I fully agreed with the rational underlying the decision to better focused on W-t-H ratio, BMI categories remain of interest to better represent the studied population. Authors may consider adding a percentage of participants with obesity, overweight or lean for each cluster. This appears to me particularly needed since data are coming from studies in obesity field. 
Line 140 : I was wondering the rational underlying the fact that authors didn’t consider Anxiety and Depression underscore of the HADS? This could be informative regarding their specific impact on food patterns, HPA axis and/or neurocognitive results?

Line 143 : If possible, it would be of interest to better develop the choice of the outcome and to highlight the executive function assess by the outcome ? Flexibility, attention, inhibitory control? This may help the reader to interpret the results
Line 163 : Authors may provide here (or elsewhere) information regarding the subpopulation that get MRI (why only 60/108 and 106/175). I also suggest adding, in supplemental material, “MRI population” version of Table 1 and Table 2.  
Line 191 : please provide the version number of the package and R
Line 240 : If possible, authors should provide the results of Executive functions test, in supplemental material. This could help for further work as meta-analysis.
Line 274 : idem line 240
Line 307 : Some will argue that the lack of results could also be due to a lack a sensitivity of cognitive tests or a lack of specificity ( food or emotional cueing? Fasting state?). Maybe this should be added here or in a limitation section.
Line 315 : Even if I agree with the author’s statement, this affirmation appears to be far away from the study results. Numerous other factors could influence. This might be balanced ?

Author Response

We thank the Reviewer for their positive comments and for their time reviewing our manuscript. Below, we provide a description of the changes implemented.

-    Due to the use of data from studies related to obesity, some confusion may be present. If possible, authors may consider to better describe BMI, BMI ranges, percentage for each BMI categories?

RESPONSE:

This information is now added in Tables 1 and 2. In the case of adolescents, raw BMI scores were not directly used to classify participants into lean, overweight, and obese categories. Rather, we applied the international cut-off points established by Cole et al. 2000, which also take into account age and sex in the classification.

-    Maybe sex and gender are under-explored, however, dietary patterns and cognitive functions are likely to be influenced by these variables. Perhaps for this study or a future one, gender-based or sex-based analysis could be considered?

RESPONSE:

We agree that sex/gender might have an influence on eating behaviors and their neurocognitive correlates. In response to this point, we have proposed this issue as a future line of research in the Discussion.

TEXT ADDED:

Sex differences might have an influence on the neurobehavioral correlates of eating patterns. In fact, pathological eating patterns seem to affect females and males differently. For instance, the prevalence of anorexia nervosa and bulimia nervosa is much higher in females than in males (e.g., [1]). While the current sample sizes were not robust enough to test for sex effects, future studies could examine if sex has an influence on the cognitive and personality profile of participants showing high scores in restrictive or uncontrolled eating.

Other comments : 
Line 31 :  maybe specify that the prevalence here is for US sample. This can differ in other studies.

RESPONSE:

We have added this specification. Now it reads: “Eating disorders, such as anorexia nervosa, bulimia nervosa, and binge eating disorder, are common mental health problems. For instance, in the United States, their life-time prevalence is estimated to be around 0.80%, 0.28%, and 0.85%, respectively [1].”

Line 53-73 : the overall paragraph is well written and of interest. However, maybe authors could add that the three circuits could be altered in the different food patterns (maybe at different levels) (e.g., anorexia nervosa with food addiction) 

RESPONSE:

We have added this point in the introduction

TEXT ADDED:

Potentially, these three circuits could be involved in different manifestations of a given eating disorder (for example, in the case of patients with the binge purge subtype of anorexia nervosa, the three circuits could hypothetically play a role in different symptoms).

Line 113-116:  is there any exclusion criteria related to MRI technic? 

RESPONSE:

We apologize for forgetting to mention it, it is now added in the main manuscript. Being claustrophobic and being bearer of non-removable metallic objects were additional exclusion criteria for the MRI technic. For other criteria, such as having tattoos, we consulted the MRI technicians and/or the radiologist and a decision was taken case-by-case (i.e., depending on the location, the size, if the ink was water-based, etc).

Line 135 : Even if I fully agreed with the rational underlying the decision to better focused on W-t-H ratio, BMI categories remain of interest to better represent the studied population. Authors may consider adding a percentage of participants with obesity, overweight or lean for each cluster. This appears to me particularly needed since data are coming from studies in obesity field. 

RESPONSE:

We fully agree. This information is now displayed in Tables 1 and 2.

Line 140 : I was wondering the rational underlying the fact that authors didn’t consider Anxiety and Depression underscore of the HADS? This could be informative regarding their specific impact on food patterns, HPA axis and/or neurocognitive results?

RESPONSE

Anxiety and depression subscores were moderately correlated with each other (r=0.45 in the adolescent sample, r=0.50 in the adult sample). We used the total score of the HADS (the sum of anxiety plus depression scores) in order to improve the score variability and in order to reduce the number of covariates added in the model. These two reasons were especially relevant in the case of the adolescent sample, since its sample size is substantially smaller than the adult sample.

We have now checked that our results in the adolescent sample remain significant when adding the two subscores as separate covariates. That is, in the following model:

IFG ~ fit.cluster + Waist_to_Height + age + sex + HADS_Anx + HADS_Dep

cluster differences in the IFG in adolescents remain significant (F value =8.713, Bonferroni-corrected p =0.013).

We have also addressed this issue as a future research line in the Discussion paragraph.

TEXT ADDED:

(…) Additionally, it would be interesting to investigate how changes in affective symptoms (such as anxiety and depression) might appear before or after certain eating behaviors, and to evaluate the neurocognitive correlates of affective symptomatology in clinical and subclinical eating disorders.

Line 143 : If possible, it would be of interest to better develop the choice of the outcome and to highlight the executive function assess by the outcome ? Flexibility, attention, inhibitory control? This may help the reader to interpret the results

RESPONSE: We have added this information in the Methods as requested

TEXT ADDED:

We used the following neuropsychological tests to examine executive function performance. We administered a computerized version of the Wisconsin card sorting test (WCST) [2] and recorded the total number of errors, and the Trail Making Test part B minus part A [3]. These tests are generally considered to measure cognitive flexibility [4]. To assess working memory [4], we administered letter-number sequencing from the Wechsler Intelligence Scales (WISC or WAIS). Finally, to evaluate inhibitory control [4], we recorded the interference score from the Stroop test [5], and number of commission errors from the Continuous Performance Test 2nd Edition (CPT-II) [6].

Line 163 : Authors may provide here (or elsewhere) information regarding the subpopulation that get MRI (why only 60/108 and 106/175). I also suggest adding, in supplemental material, “MRI population” version of Table 1 and Table 2.

We have added this information in the Methods section as follows.

Text added:

Participants were asked to participate in a brain MRI acquisition, performed on a separate day. A number of participants did not perform this session, which reduced the adolescent MRI subsample to n=60 and the adult MRI subsample to n=106. The main reasons why participants dropped out of the MRI substudy were schedule incompatibilities and meeting exclusion criteria for the MRI (see paragraph 2.1.).

As suggested, we have also added two Tables in the supplementary material displaying the clinical characteristics of the MRI subsamples (see Tables S2 and S4).

Line 191 : please provide the version number of the package and R

RESPONSE: We now provide this information as follows:

Text added:

(…) using the “cluster” package (Version 2.1.0.) available in R (Version 4.0.0.).

Line 240 : If possible, authors should provide the results of Executive functions test, in supplemental material. This could help for further work as meta-analysis.

Line 274 : idem line 240

In response to these two comments, the Supplementary material now shows the results of the executive function tests (Tables S1 and S3).

Line 307 : Some will argue that the lack of results could also be due to a lack a sensitivity of cognitive tests or a lack of specificity ( food or emotional cueing? Fasting state?). Maybe this should be added here or in a limitation section.

We have added this point as a limitation and as a future direction as follows:

Text added:

“(…) First, the neuropsychological tests administered here were standard cognitive tests that have been widely used to evaluate executive functions [4]. However, these tests might not necessarily detect differences in executive functions that are specifically linked with food stimuli. For instance, participants with uncontrolled eating might show lower inhibitory control in response to food items, but that this might not necessarily translate to their performance in the classic Stroop or CPT-II tests. For this reason, the inclusion of food-related measures of executive functions would have been desirable and it is a future line of research.”

Line 315 : Even if I agree with the author’s statement, this affirmation appears to be far away from the study results. Numerous other factors could influence. This might be balanced ?

RESPONSE: We were not completely sure that we are all referring to the same line. In the original manuscript, line 315, Discussion, it reads:

“In adolescents, there was a group effect of cortical thickness in the inferior frontal gyrus, in which participants in the restrained eating group showed lower thickness than both the uncontrolled eating and the low problematic eating groups.”

We have left these lines unmodified since, in our opinion, they are merely reflecting the results of the ANCOVA and the post-hoc tests.

We believe that the reviewer might be referring to line 351, Discussion, 2 paragraphs later, which in the original manuscript, it read:

“Brain cortical development during adolescence is characterized by an accelerated thinning with increasing age, which seems to be especially prominent in frontal lobe regions [7]. A possible explanation for our findings is that high scores in restrictive eating might alter normative trajectories of neural development in the inferior frontal gyrus, a region associated with general inhibitory processes [8] and more specifically, with dietary self-control [9].”

We have modified this statement as follows:

“Brain cortical development during adolescence is characterized by an accelerated thinning with increasing age, which seems to be especially prominent in frontal lobe regions [7]. Hypothetically speaking, restrictive eating behaviors could interact with the neural development in the inferior frontal gyrus, a region associated with general inhibitory processes [8] and more specifically, with dietary self-control [9]. This is purely speculative, since we would need longitudinal designs to trace this interaction.”

If none of this was the case and we have misunderstood the reviewer, we deeply apologize for this and we ask for another opportunity to implement the changes.